# Co-Application of Statin and Flavonoids as an Effective Strategy to Reduce the Activity of Voltage-Gated Potassium Channels Kv1.3 and Induce Apoptosis in Human Leukemic T Cell Line Jurkat

**DOI:** 10.3390/molecules27103227

**Published:** 2022-05-18

**Authors:** Andrzej Teisseyre, Mateusz Chmielarz, Anna Uryga, Kamila Środa-Pomianek, Anna Palko-Łabuz

**Affiliations:** 1Department of Biophysics and Neurobiology, Wrocław Medical University, Chałubińskiego 3A, 50-368 Wrocław, Poland; andrrzej.teisseyre@umw.edu.pl (A.T.); mateusz.chmielarz@student.umw.edu.pl (M.C.); anna.uryga@umw.edu.pl (A.U.); 2Department of Microbiology, Wrocław Medical University, Chałubińskiego 4, 50-368 Wrocław, Poland

**Keywords:** Kv1.3 channel, Jurkat T cell, statin, flavonoid, cancer apoptosis, cancer therapy

## Abstract

Voltage-gated potassium channels of the Kv1.3 type are considered a potential new molecular target in several pathologies, including some cancer disorders and COVID-19. Lipophilic non-toxic organic inhibitors of Kv1.3 channels, such as statins and flavonoids, may have clinical applications in supporting the therapy of some cancer diseases, such as breast, pancreas, and lung cancer; melanoma; or chronic lymphocytic leukemia. This study focuses on the influence of the co-application of statins—simvastatin (SIM) or mevastatin (MEV)—with flavonoids 8-prenylnaringenin (8-PN), 6-prenylnarigenin (6-PN), xanthohumol (XANT), acacetin (ACAC), or chrysin on the activity of Kv1.3 channels, viability, and the apoptosis of cancer cells in the human T cell line Jurkat. We showed that the inhibitory effect of co-application of the statins with flavonoids was significantly more potent than the effects exerted by each compound applied alone. Combinations of simvastatin with chrysin, as well as mevastatin with 8-prenylnaringenin, seem to be the most promising. We also found that these results correlate with an increased ability of the statin–flavonoid combination to reduce viability and induce apoptosis in cancer cells compared to single compounds. Our findings suggest that the co-application of statins and flavonoids at low concentrations may increase the effectiveness and safety of cancer therapy. Thus, the simultaneous application of statins and flavonoids may be a new and promising anticancer strategy.

## 1. Introduction

Voltage-gated potassium channels of the Kv1.3 type are widely expressed in many types of cells, both in the plasma membrane and in the inner mitochondrial membrane (mito Kv1.3 channels) [1,2,3,4]. The activity of Kv1.3 channels is required, among others, in cell proliferation and apoptosis [2,3,4,5]. The specific inhibitors of Kv1.3 channels may potentially be applied in the treatment of T-lymphocyte-mediated autoimmune diseases [4,5] and chronic respiratory diseases, including severe cases of COVID-19 disease [6,7]. Since the expression of Kv1.3 channels is up-regulated in some cancer types, such as breast and lung cancer, melanoma, pancreatic ductal adenocarcinoma, or chronic lymphocytic leukemia (B-CLL) [4,8,9,10], inhibitors of the channels may also clinical applications in supporting the chemotherapy of these types of cancer [9,10,11]. Among inhibitors of Kv1.3 channels, the most promising candidates for potential clinical application in cancer therapy are lipophilic small-molecule organic compounds that are able to induce selective cancer cell apoptosis (by the inhibition of mito Kv1.3 channels) while sparing normal cells [9,10]. Also included in this group are some plant-derived flavonoids, chalcones, and substituted stilbenes that were tested in our laboratory [9,10].

Recently published results showed that the ability to inhibit Kv1.3 channels is shared by three plant-derived statins: pravastatin, mevastatin, and simvastatin [12]. Statins are organic compounds that are applied in the treatment of hypercholesterolemia and atherosclerosis [13] and might also be key therapeutic agents in the therapy of severe COVID-19 cases [14]. Moreover, the statins evastatin and simvastatin exert antiproliferative, pro-apoptotic, and drug-resistance reversal effects in Kv1.3 channel-expressing human colon adenocarcinoma cell line LoVo and its doxorubicin-resistant subline LoVo/Dx [15]. Interestingly, the inhibitory effect of simvastatin and mevastatin on Kv1.3 channels resembles the effect exerted on the channels by the flavonoids 8-prenylnaringenin (8-PN), acacetin, and chrysin [10,12]. This may indicate that the inhibitory effects exerted on Kv1.3 channels by statins and flavonoids are additive. Additivity of inhibitory effects was observed in the case of co-application of the isoflavone genistein and a substituted stilbene resveratrol, which inhibited Kv1.3 channels in a very similar manner [16]. The putative additivity of the inhibitory effects of statins and flavonoids on Kv1.3 channels may be related to the synergistic antiproliferative and pro-apoptotic effects of statins co-applied with flavonoids on Kv1.3 channel-expressing cancer cells. Such a synergism was observed upon co-application of simvastatin and mevastatin with flavonoids 6-hydroxyflavone and baicalein [15] and upon co-application of atorvastatin with nobiletin and phloretin [17,18]. The synergistic anticancer activity of statins co-applied with flavonoids is of importance from the point of view of the possible application of these compounds in cancer therapy. Such a synergism may allow a reduction in required therapeutic doses. This may reduce the probability of unwanted side effects of the drugs due to the cytotoxicity of statins.

This study focuses on the influence of the co-application of simvastatin or mevastatin with 8-prenylnaringenin, 6-prenylnarigenin, xanthohumol, acacetin, or chrysin on the activity of Kv1.3 channels, viability, and apoptosis of cancer cells in the human T cell line Jurkat. Human leukemic T cell line Jurkat, which endogenously expresses a large number of Kv1.3 channels [19,20], was used in our study as a model system of cancer cells. All tested compounds reversibly inhibited Kv1.3 channels in cancer cells (Jurkat T) when applied alone at micromolar concentrations [21,22,23]. Results provide evidence that the inhibitory effects of simvastatin co-applied with mevastatin are not additive. However, clear additivity was observed upon co-application of both statins with 8-prenylnaringenin, 6-prenylnaringenin, and chrysin and upon co-application of mevastatin with xanthohumol and acacetin. It was found that these results correlate with an increased ability of the statin–flavonoid combination to reduce viability and induce apoptosis in studied cancer cells compared to single compounds. To summarize, the ability of the studied compounds to inhibit Kv1.3 channels in Jurkat T cells may be correlated with the ability to reduce viability and induce apoptosis of these cells.

## 2. Results

### 2.1. Electrophysiological Studies

Figure 1A depicts whole-cell currents recorded in a Jurkat T cell applying a “voltage ramp” (see Materials and Methods) under control conditions, upon a co-application of simvastatin and mevastatin at the concentration of 6.0 μM, and after wash-out of the compounds. The concentration of 6.0 μM was chosen because it was shown that the statins applied at this concentration significantly and reversibly inhibited whole-ell Kv1.3 currents in Jurkat T cells [12].

The figure depicts raw currents containing two components: small linear and much bigger non-linear currents. The linear current was the unspecific leak current, not interesting for our study, whereas the non-linear one was the actual Kv1.3 channel current [24]. This current was significantly diminished upon co-application of the statins. The current recovered completely after the wash-out of the compounds. This indicates that the inhibitory effect of the statins on the channels is reversible.

Figure 1B depicts relative peak currents (see Materials and Methods) calculated upon co-application of the statins and each of the statins alone. The currents were equal to 0.47 ± 0.14 (*n* = 28), 0.50 ± 0.09 (*n* = 16), and 0.41 ± 0.11 (*n* = 21) of the control value upon co-application of simvastatin and mevastatin and the application of simvastatin alone and mevastatin alone, respectively. The currents recorded upon a co-application of simvastatin and mevastatin and in the presence of simvastatin or mevastatin applied alone did not significantly differ one from another (*p* > 0.05, one-way ANOVA). Thus, a co-application of simvastatin and mevastatin does not influence the magnitude of their inhibitory effect on Kv1.3 channels.

It is known that the inhibitory effects of both studied statins are accompanied by a significant acceleration of the inactivation rate of Kv1.3 currents by means of a significant reduction in the inactivation time constant [12]. Therefore, it was of interest to prove whether a co-application of simvastatin and mevastatin could further influence the inactivation rate of the currents. Figure 1C shows normalized whole-cell potassium currents recorded applying depolarizing voltage steps (see in Materials and Methods) under control conditions upon the co-application of simvastatin and mevastatin (both applied at a concentration of 6.0 μM) and after the wash-out of the compounds. Apparently, a co-application of simvastatin and mevastatin accelerated the inactivation of the currents. The current did not recover completely after the wash-out of the statins. This is in agreement with our earlier results, which showed that acceleration of the inactivation by simvastatin was partially irreversible [12].

Figure 1D depicts values of inactivation time constants upon co-application of the statins and each of the statins alone at the same concentration. The values were equal to 18.19 ± 3.97 ms (*n* = 20), 16.95 ± 4.2 ms (*n* = 31), and 24.56 ± 11.57 ms (*n* = 11) upon the co-application of simvastatin and mevastatin and upon the application of simvastatin and mevastatin alone, respectively.The inactivation time constants were not significantly different one from another (*p* > 0.05, one-way ANOVA). Thus, the co-application of simvastatin and mevastatin had no influence on the inactivation rate of Kv1.3 currents.

Figure 2A depicts relative peak currents upon co-application of simvastatin with 8-prenylnaringenin, simvastatin applied alone, and 8-prenylnaringenin applied alone, respectively, whereas Figure 2B shows the same data obtained upon co-application of mevastatin with 8-prenylnaringenin, mevastatin applied alone, and 8-prenylnaringenin applied alone. The concentration of 8-prenylnaringenin was chosen to be 3 μM because such a concentration was high enough to inhibit most of the Kv1.3 channels in Jurkat T cells [21]. The concentrations of the statins were equal to 6 μM. The relative peak current upon co-application of simvastatin and 8-prenylnaringenin was equal to 0.12 ± 0.07 (*n* = 15). This value was significantly smaller than the values obtained for simvastatin (see above) and 8-prenylnaringenin applied alone (0.39 ± 0.10, (*n* = 15), *p* < 0.05, one-way ANOVA). The inactivation time constant for simvastatin co-applied with 8-prenylnaringenin was equal to 13.89 ± 4.39 (*n* = 12). This value was not significantly different from the values calculated for simvastatin applied alone (see above, *p* > 0.05, one-way ANOVA), but it was significantly smaller than the values obtained for 8-prenylnaringenin applied alone (35.52 ± 8.99 (*n* = 13), *p* < 0.05, one-way ANOVA).

The relative peak current for mevastatin co-applied with 8-prenylnaringenin was equal to 0.09 ± 0.064 (*n* = 19). This value was significantly smaller than the values obtained for mevastatin and 8-prenylnaringenin applied alone (see above, *p* < 0.05, one-way ANOVA). The inactivation time constant upon co-application of simvastatin and mevastatin with 8-prenylnaringenin was not significantly different from the value calculated for simvastatin and mevastatin applied alone (data not shown).

Figure 2C depicts relative peak currents upon co-application of simvastatin with 6-prenylnaringenin, simvastatin applied alone, and 6-prenylnaringenin applied alone, respectively, whereas Figure 2D shows the same parameter obtained upon co-application of mevastatin with 6-prenylnaringenin, mevastatin applied alone, and 6-prenylnaringenin applied alone. The concentration of 6-prenylnaringenin was equal to 6.0 μM because such a concentration was high enough to significantly inhibit Kv1.3 channels in Jurkat T cells [19]. The concentrations of the statins were previously mentioned. The relative peak current for simvastatin co-applied with 6-prenylnaringenin was equal to 0.26 ± 0.17 (*n* = 18). This value was significantly smaller than the values obtained for simvastatin (see above) and 6-prenylnaringenin applied alone (0.79 ± 0.09 (*n* = 11), *p* < 0.05, one-way ANOVA). The inactivation time constant for simvastatin co-applied with 6-prenylnaringenin was equal to 18.9 ± 6.31 (*n* = 14). This value was not significantly different from the values calculated for simvastatin applied alone (see above, *p* > 0.05, one-way ANOVA), but it was significantly smaller than the value obtained for 6-prenylnaringenin applied alone (68.33 ± 20.74 (*n* = 3), *p* < 0.05, one-way ANOVA).

The relative peak current for mevastatin co-applied with 6-prenylnaringenin was equal to 0.29 ± 0.069 (*n* = 20). This current was significantly smaller than the currents obtained for mevastatin and 6-prenylnaringenin applied alone (see above, *p* < 0.05, one-way ANOVA). The inactivation time constant for simvastatin and mevastatin co-applied with 6-prenylnaringenin was not significantly different from the value calculated for simvastatin and mevastatin applied alone (data not shown).

Figure 2E depicts the relative peak currents upon co-application of mevastatin with xanthohumol, mevastatin applied alone, and xanthohumol applied alone. The concentration of xanthohumol was equal to 6.0 μM because such a concentration was high enough to significantly inhibit Kv1.3 channels in Jurkat T cells [20]. The relative peak current for mevastatin co-applied with xanthohumol was equal to 0.30 ± 0.11 (*n* = 25). This current was significantly smaller than the currents obtained for mevastatin (see above) and xanthohumol applied alone (0.72 ± 0.14, (*n* = 10), *p* < 0.05, one-way ANOVA). The relative peak current upon co-application of simvastatin with xanthohumol was equal to the value calculated for simvastatin applied alone (not shown). Since the application of xanthohumol did not change the inactivation rate of Kv1.3 currents [20], the effect of co-application of simvastatin or mevastatin with xanthohumol on the inactivation rate of the currents was not studied in detail.

Figure 2F depicts relative peak currents upon co-application of mevastatin with acacetin, mevastatin applied alone, and acacetin applied alone. The concentration of acacetin was equal to 30.0 μM because such a concentration was high enough to significantly inhibit Kv1.3 channels in Jurkat T cells [19]. The concentration of the statins was previously mentioned. The relative peak current for mevastatin co-applied with acacetin was equal to 0.30 ± 0.09 (*n* = 31). This value was significantly smaller than the values calculated for mevastatin (see above, *p* < 0.05, one-way ANOVA) and acacetin applied alone (0.64 ± 0.14 (*n* = 19), *p* < 0.05, one-way ANOVA). The relative peak current upon the co-application of simvastatin with acacetin was not significantly smaller than the value for simvastatin applied alone (not shown).

Figure 2G depicts relative peak currents upon co-application of simvastatin with chrysin, simvastatin applied alone, and chrysin applied alone, respectively, whereas Figure 2H shows the same data calculated upon co-application of mevastatin with chrysin, mevastatin applied alone, and chrysin applied alone. The concentration of chrysin was equal to 30.0 μM because such a concentration was high enough to significantly inhibit Kv1.3 channels in Jurkat T cells [19]. The concentration of the statins was previously mentioned. The relative peak current upon co-application of simvastatin with chrysin was equal to 0.25 ± 0.11 (*n* = 21). This value was significantly smaller than the values calculated for simvastatin (see above, *p* < 0.05, one-way ANOVA) and chrysin applied alone (0.57 ± 0.13 (*n* = 18), *p* < 0.05, one-way ANOVA). The inactivation time constant upon co-application of simvastatin with chrysin was equal to 20.64 ± 5.16 (*n* = 9). This value was not significantly different from the value calculated for simvastatin applied alone (*p* > 0.05, one-way ANOVA), but it was significantly smaller than the value obtained for chrysin applied alone (34.7 ± 15.46 (*n* = 9), *p* < 0.05, one-way ANOVA). The relative peak currents for mevastatin co-applied with chrysin were equal to 0.26 ± 0.11 (*n* = 21). This value was significantly smaller than the values calculated for mevastatin (see above, *p* < 0.05, one-way ANOVA) and chrysin applied alone (*p* < 0.05, one-way ANOVA). The inactivation time constant for simvastatin and mevastatin co-applied with chrysin was not significantly different from the value calculated for simvastatin and mevastatin applied alone (not shown).

The fact that the magnitude of channel inhibition upon co-application of the statins with selected flavonoids is significantly stronger than the effect exerted of each compound applied alone may be a consequence of the additivity of their inhibitory effects. If two channel inhibitors act on the same molecular target independently of each other, their inhibitory effects may be additive. In such a case, the relative peak current upon co-application of the inhibitors is equal to the product of multiplication of the relative currents upon application of each of the inhibitors alone. Table 1 shows the relative peak currents upon co-application of the statins with the selected flavonoids, calculated applying the additive inhibitory effect model, compared to the values obtained under experimental conditions. Statistical significance between theoretical and experimental values was estimated by applying Student’s unpaired *t*-test. The difference is considered to be statistically significant when *p* < 0.05. The term significant (+) means that the inhibitory effect is significantly stronger than it is predicted by the additive inhibition model, whereas the term significant (−) means that it is significantly weaker.

Table 1 shows that the magnitude of the inhibitory effect upon co-application of simvastatin with chrysin and mevastatin with all the compounds except for 8-prenylnaringenin was not significantly different from the magnitude predicted by the additive inhibition model. However, in the case of both statins co-applied with 8-prenylnaringenin and simvastatin co-applied with 6-prenylnaringenin, the inhibitory effect was significantly stronger than it was predicted by the additive inhibition model. On the other hand, in the case of simvastatin co-applied with xanthohumol and acacetin, the inhibitory effect was significantly weaker than predicted by the theoretical model.

### 2.2. Cell Viability in the Presence of Studied Compounds

Simvastatin and mevastatin exhibited a dose-dependent ability to inhibit the growth of Jurkat T cells. The concentration of mevastatin that corresponds to half of its maximal inhibitory effect (IC_50_) was 22.5 µM. In the case of simvastatin, it was not possible to determine the IC_50_ value in the studied concentration range. However, the lowest cell survival in the presence of the statin was about 60% at a concentration of 40.0 µM (Figure 3).

The ability of most flavonoids (6-prenylnaringenin, chrysin, and acacetin) to inhibit Jurkat T cells growth was shown in the work of Teisseyre et al. (2018). 6-prenylnaringenin and acacetin exhibited low potency to reduce the cells’ growth (IC_50_ not available), whereas IC_50_ = 26.2 µM was noted for chrysin [19]. In the present studies, we also determined cell viability in the presence of 8-prenylnaringenin and xanthohumol. We found that 8-prenylnaringenin, similar to the 6-prenylated derivative, is characterized by low activity as a growth inhibitor. However, the presence of the prenyl group in the structure of xanthohumol, which has an open C-ring with a hydroxyl group on the B-ring, may contribute to increasing the ability of the compound to kill Jurkat cells (IC_50_ = 32.5 µM) (Table 2). The potency of statins, simvastatin, and mevastatin to increase the activity of flavonoids was investigated. In the presence of mevastatin, we observed a much lower value of IC50 for all studied flavonoids, except acacetin (IC_50_ not available). Simvastatin and mevastatin increased cell growth inhibition by prenylated flavanones and chrysin (Table 2). The most significant changes between the activity of pure flavonoid and its combinations with statins were observed in the case of 8-prenylnaringenin (Figure 4A). However, the highest potency of Jurkat T cells growth inhibition was noticed for a combination of xanthohumol and mevastatin (Figure 4B).

### 2.3. Apoptosis Induction

Both studied statins induced the activity of caspase-3 (*p* < 0,05); however, the potency of mevastatin was more significant than simvastatin (Figure 5A). In the case of single flavonoids, only acacetin (Figure 5B) and chrysin (Figure 5C) increased the activity of the studied enzyme. However, the combination use of acacetin and statins did not change the effect obtained for flavonoids only (Figure 5B). It is worth emphasizing that an increase in caspase-3 activity was noted with the simultaneous application of mevastatin with other flavonoids as opposed to their solitary use. The most significant changes were observed when mevastatin was administered in combination with 8-prenylnaringenin. The addition of simvastatin to prenylated derivatives also enhanced the induction of caspase-3 activity. However, we did not observe any statistically significant differences between the activity of this enzyme in the presence of other flavonoids and its activity in the presence of their combinations with simvastatin (Figure 5D–F).

Caspase-3 is one specific effector caspase, a protein that is cleaved and thus activated upon the initiation of apoptosis. Cleaved caspase-3 propagates an apoptotic signal through enzymatic activity on downstream targets. Both uncleaved and cleaved versions of the enzyme are strong indicators of cell death induction. In our study, cleaved forms of caspase-3 increased upon co-application of the drugs, which was demonstrated using the Western blot technique (Figure 6).

In our studies, the effect of the studied compound on the mitochondrial membrane potential (MMP) was determined. MMP was expressed as a fraction of cells with green fluorescence (490 nm/525 nm) in relation to the fraction of cells with red fluorescence (540 nm/590 nm), which corresponded to the monomers and aggregates’ fluorescence, respectively. We found that both statins increase the fraction of JC-10 monomers. Moreover, the effect noted in the presence of mevastatin was more significant compared to simvastatin. (Figure 7A). A reduction in MMP was also observed in the cells incubated with acacetin and chrysin. We observed that the activity of these flavonoids was potentiated by simvastatin as well as mevastatin (Figure 7B–C). We did not find any changes between the fraction of JC-10 monomers in control cells and in the cells cultured with xanthohumol and prenylated derivatives of naringenin. The presence of simvastatin only slightly increased the ability of these flavonoids to reduce MMP. However, statistically significant changes were only noted in the case of mevastatin (Figure 7D–F).

## 3. Discussion

The results of this study showed that the co-application of simvastatin with mevastatin did not further increase the magnitude of the inhibitory effect exerted by the statins on Kv1.3 channels in human Jurkat T cells. Our results published earlier showed that both simvastatin and mevastatin inhibited Kv1.3 channels in Jurkat T cells, probably via a complex mechanism [12]. A lack of augmentation of the channel inhibition upon co-application of simvastatin and mevastatin may suggest that both statins share the mechanism of inhibition of the channels.

In contrast, co-application of both statins with most of the flavonoids caused a significant augmentation of the inhibition of Kv1.3 channels. The magnitude of inhibition upon co-application of the statins with the flavonoids could be described by applying the additive inhibition model. This model enabled calculations of relative peak currents not significantly different from experimental values in the case of simvastatin co-applied with chrysin and mevastatin co-applied with 6-prenylnaringenin, xanthohumol, acacetin, and chrysin.

However, our data also showed that inhibition upon co-application of the statins with the flavonoids might not be a result of simple additivity of inhibitory effects of the inhibitors. First of all, the inhibition upon co-application of simvastatin and mevastatin with 8-prenylnaringenin and simvastatin with 6-prenylnaringenin was significantly more potent than predicted by the additive model. On the other hand, inhibition upon co-application of simvastatin with xanthohumol and acacetin was significantly less potent than predicted by this model. In this case, no additivity was observed. The inhibition was comparable to the inhibitory effect of simvastatin applied alone. Finally, co-application of the statins with the flavonoids did not further accelerate the inactivation rate of the channels. Such an acceleration was expected if the inhibitory effects of co-applied compounds were additive. Inactivation time constants upon co-application of the statins with the flavonoids were comparable to the values obtained for the statins applied alone, except for simvastatin and mevastatin co-applied with acacetin, where they were significantly higher. Altogether, our results demonstrate that the mechanism of the channel inhibition may be more complex, including a synergistic action of both inhibitors on the channels in the case of co-application of the statins with 8-prenylnaringenin and 6-prenylnaringenin. More studies are necessary to further elucidate this problem.

Many studies have demonstrated the cytotoxic activity of statins against breast cancer [25,26], colon cancer [27], and leukemia [28]. However, only a few reports deal with the influence of combined treatment with statins and flavonoids on cancer cell survival.

In our previous studies, 6-prenylnaringenin, chrysin, and acacetin were identified as growth inhibitors of Jurkat T cells [19]. In this study, the cytotoxic effect of two additional flavonoid derivatives, 8-prenylnaringenin and xanthohumol, was also observed. In addition, both studied statins exerted a dose-dependent cytotoxic effect on the studied cancer cells. The most important result was the observation that co-treatment of Jurkat T cells with the flavonoids and the statins caused a reduction in the concentration needed to produce a 50% reduction in cell growth (IC_50_). This may suggest that the statins may act as effective amplifiers of flavonoid activity as growth inhibitors of cancer cells. The most significant, almost 10-fold, was the reduction in IC50 in the case of xanthohumol co-applied with mevastatin. On the other hand, no significant reduction in the IC50 value was observed upon co-application of xanthohumol with simvastatin. In the case of both prenylated derivatives of naringenin and chrysin, the IC50 value upon co-application of the compounds with mevastatin was lower than the value upon the co-application with simvastatin. This may suggest that a co-application of the flavonoids with mevastatin is more effective than co-application with simvastatin. This may be due to the fact that mevastatin appeared to be a more effective inhibitor of the growth of Jurkat T cells than simvastatin.

Anticancer activity of different compounds usually results from their ability to induce apoptosis of cancer cells. Apoptosis is a stringently organized process regulated by a series of signal transduction cascades and cellular proteins [29]. Pro-apoptotic proteins are classified into Caspases and BCL-2 families. Caspases are divided into three subtypes: inflammatory caspases; initiator caspases, in intrinsic and extrinsic apoptotic pathways; and effector caspases. Depending on the type of cells, statins, as well as flavonoids, can induce different molecular pathways initiating apoptosis [30]. Apoptosis involves a disruption of mitochondrial membrane integrity that is decisive for the cell death process. Therefore, in this study, the apoptotic effects of studied compounds on Jurkat T cells were evaluated by three methods: caspase-3 activity, caspase-3 Western blot analysis, and loss of mitochondrial membrane potential (MMP). All results of the apoptotic assays indicate that Jurkat T cells went along the apoptotic pathway after mevastatin and simvastatin treatment. The potency of the mevastatin was more significant than that of simvastatin, as shown by means of an increase in caspase-3 activity and loss of MMP, in addition to the presence of caspase-3 fragments. In our previous research, we have shown that statins applied alone and together with flavonoids induce apoptosis in sensitive and resistant to doxorubicin colorectal cancer cells [15]. In this study on Jurkat T cells, statins were also co-applied with the selected flavonoids. Our results showed a correlation with studies by Watanabe et al. [31]. Similarly, we also found that inhibition of the growth of Jurkat T cells by acacetin and chrysin occurs due to the activation of caspase-3. This observation was also reflected in the reduced value of the MMP upon application of these compounds.

It is known that antiproliferative and pro-apoptotic effects of the statins on Kv1.3-channel expressing cancer cells may be related to the inhibition of Kv1.3 channels [12,15]. Results of this study may indicate that ability of statins and the statins co-applied with flavonoids to inhibit cancer cell growth and to induce the apoptosis of these cells may be related to the effectiveness of inhibition of Kv1.3 channels in these cells. The ability of statins and the statins co-applied with the flavonoids, except for acacetin, to inhibit growth and induce apoptosis of Jurkat T cells correlates with their ability to inhibit Kv1.3 channels in these cells. For example, it was shown that the most significant inhibition of cancer cell growth and induction of apoptosis occurs in the case of co-application of statins with prenylated derivatives of naringenin and mevastatin co-applied with xanthohumol. This correlates with these compounds’ ability to inhibit the channels upon co-application, which is the most significant in the case of the co-application of statins with prenylated derivatives of naringenin and mevastatin co-applied with xanthohumol. On the other hand, upon co-application of simvastatin and xanthohumol, no significant change in the inhibition of Kv1.3 channels was observed, and this correlates with a lack of effect of co-application on cancer cell growth and apoptosis.

As mentioned earlier, because of their ability to inhibit Kv1.3 channels and lipophilicity, statins may potentially have clinical applications in supporting the chemotherapy of some cancer types [10,12]. In the area of clinical application, a very important question is the question of the required dose. The dose should be as low as possible due to possible unwanted side effects. Reducing the required dose can be achieved by the co-application of two drugs that exert synergistic or additive effects on their molecular target when applied together. In such a case, satisfactory therapeutic results can be obtained with both drugs applied at low concentrations. A simultaneous application of statins and flavonoids may reduce the required therapeutic dose.

Another problem that has to be resolved while considering the putative application of the tested compounds in medicinal practice is the question of their bioavailability. Cellular and molecular mechanisms underlying therapeutic applications of chrysin [32], xanthohumol, acacetin [33], and 8-prenylnaringenin in various cancers have been gathered and discussed in many papers. However, the low bioavailability of flavonoids such as chrysin [34] due to rapid metabolism and excretion causes its use to be less beneficial when compared to other anticancer components from natural plants [35]. This problem may also be resolved by combination therapies, which entail the combined use of two cancer cell growth inhibitors. The simultaneous use of statins and flavonoids applied at low concentrations can increase their effectiveness and the safety of therapy.

## 4. Material and Methods

### 4.1. Chemicals

Simvastatin (SIM) and mevastatin (MEV) were purchased from Sigma-Aldrich (St. Louis, MI, USA). All flavonoids (acacetin, ACAC; chrysin, CHR;8-prenylnaringenin, 8-PR; 6-prenylnaringenin, 6-PR) and xanthohumol (Xant) were purchased from Alexis Biochemicals. Their chemical structures are presented in Figure 8. Stock solutions of all compounds were prepared in dimethyl sulfoxide (DMSO).

### 4.2. Cell Culture and Solutions

Human leukemic T cell line, Jurkat (clone E6-1), was purchased from American Type Culture Collection (Manassas, VA, USA). The cells were grown in RPMI 1640 medium (Sigma-Aldrich, St. Louis, MO, USA) containing 10% heat-inactivated FBS, 10 mM HEPES, and 2.0 mM glutamate. Cells were grown on culture plates at 37 °C in a 5% CO_2_-humidified incubator. During patch-clamp experiments cells were washed by the external solution containing 150.0 mM NaCl, 4.5 mM KCl, 1.0 mM CaCl_2_, 1.0 mM MgCl_2_, 10.0 mM HEPES: pH of solution = 7.35, adjusted with NaOH. The internal (pipette) solution used for whole-cell recording contained 150.0 mM KCl, 1 mM CaCl_2_, 2.0 mM MgCl_2_, 10.0 mM HEPES, 10.0 mM EGTA, pH of solution = 7.2, adjusted with KOH. The concentration of free calcium ions in the internal solution was below 100.0 nM in order to prevent activation of calcium-activated K+ channels KCa2.2, which are expressed in Jurkat T cells [36,37]. The chemicals used in preparation of the solutions were purchased from the Polish Chemical Company (POCH, Gliwice, Poland), except for HEPES and EGTA, which were purchased from Sigma-Aldrich. The examined compounds (statins and flavonoids) were purchased from Alexis Biochemicals (Lausen, Switzerland).

### 4.3. Patch-Clamp Recordings

Dishes with examined Jurkat T cells were placed under an inverted Olympus IMT-2 microscope. Washing solutions and solutions containing examined statins and flavonoids were applied using eight-channel gravitation perfusion system (ALA Scientific Instruments, Farmingdale, NY, USA). Pipettes were pulled from a borosilicate glass (Hilgenberg, Germany). The pipette resistance was higher than 3.0 MΩ.

Kv1.3 currents in Jurkat T cells were recorded applying the whole-cell patch-clamp technique [38]. The currents were recorded using an EPC-7 Amplifier (HEKA, Germany), low-pass filtered at 3.0 kHz, and digitised using a CED Micro 1401 analog-to-digital converter (Cambridge, UK) with a sampling rate of 10.0 kHz. The influence of selected compounds on the activity of the channels was studied by applying the voltage ramp protocol. During application of a voltage ramp, the membrane of examined cell was gradually depolarised from −100.0 mV up to +40.0 mV. The ramp duration was 340.0 ms. The ramps were applied every 30.0 s, and the holding membrane potential kept at the meantime was −90.0 mV [12]. Upon application of the voltage ramp protocol, potassium currents in Jurkat T cells could be stably recorded for at least 20 min after “break-in” to the whole-cell configuration. During the off-line analysis, the leak current estimated at +40.0 mV was subtracted from the total ramp current recorded at this voltage, giving the value of peak Kv1.3 currents.

In order to study the influence of selected compounds on the channel inactivation kinetics in more detail, another protocol containing a sequence of depolarizing voltage steps from the holding potential of −90.0 mV to +60.0 mV (500.0 ms step duration) was applied every 30.0 s [12]. All experiments were carried out at room temperature (22–24 °C). The data are presented as mean ± standard deviation.

### 4.4. MTT Assay

The influence of studied compounds on the viability of Jurkat T cells was determined using MTT (3-(4,5-dimethylthiazol-2-yl)-2,5-diphenyl-2H-tetrazolium bromide) test as described previously [19]. The impact of single statins or flavonoids was examined in the concentration range of 2.5–40.0 µM. In case of compounds’ combinations, flavonoids were applied in the concentration range of 2.5–40.0 µM, whereas the concentration of statin (simvastatin or mevastatin) was constant (6.0 µM).

### 4.5. Caspase-3 Activity

The activity of caspase-3 was determined with use of Caspase-3 Colorimetric Assay Kit (GenScript), according to the manufacturer’s protocol with minor modifications. In this experiment, chromophore p-nitroaniline (pNA) released by the enzyme from the labeled substrate DEVD-pNA was detected by spectrophotometric method. Jurkat T cells (3.0 × 10^6^) were cultured in the presence of compound for 48.0 h. The cultures were maintained in humidified atmosphere of 5% CO_2_ at 37 °C. In the next step, cells were centrifuged (900.0 rpm/RT/5 min); pellet in each of probes was washed with PBS and centrifuged once again (900.0 rpm/RT/5 min). Next, supernatants were removed, and the cells were re-suspended in Lysis Buffer. The samples were incubated for 60.0 min on ice and mixed every 15.0 min. Then, mixtures were centrifuged (7000.0 rpm/4 °C/1 min) and concentration of proteins in cytosolic extract (supernatant) was determined using Bradford method. A total of 200.0 µg of protein was mixed with Reaction Buffer (containing 10.0 mM DTT) and then caspase-3 substrate (DEVD-pNA) to a final concentration of 200.0 µM was added to the samples. The reaction was performed for 4 h at a temperature of 37 °C. In the final step, the absorbance of the samples at 400 nm was measured. The relative increase in caspase-3 activity was determined by comparing pNA absorbance recorded for samples incubated with compounds and control samples. In the experiments on single compounds, we applied concentration of 30.0 µM in case of chrysin and xanthohumol, 40 µM in case of other flavonoids or simvastatin, and 20.0 µM in case of mevastatin. In the experiment on combinations of the compounds, we applied 30.0 µM (chrysin or xanthohumol) or 40 µM (prenylated derivatives or acacetin) of flavonoid and 6.0 µM of statin.

### 4.6. MMP Investigation

In order to determine the loss of mitochondrial membrane potential (MMP) Mitochondrial Membrane Potential Kit (Sigma-Aldrich, St. Louis, MI, USA) was used. The test is based on the fluorimetric detection of cationic, lipophilic JC-10 dye, which in normal cells concentrates in the mitochondrial matrix in the form of aggregates characterized by red fluorescence (λ_ex_ = 540/λ_em_ = 590 nm). In apoptotic or necrotic cells, the loss of MMP leads to decomposition of aggregates to form green fluorescent monomers (λ_ex_ = 490/λ_em_ = 525 nm). The procedure was performed according to the manufacturer’s protocol with minor modifications. In our experiments, Jurkat cells were seeded at a density of 3 × 10^4^ onto 96-well and treated with examined compounds. The cultures were maintained in humidified atmosphere of 5% CO_2_ at 37 °C. Time of incubation was 48 h. In the next step, plates were centrifuged (500.0 rpm/RT/5 min), and the medium was removed. Then, 100.0 µL of Dye Loading Solution was added to each well, and the plates were incubated 5% CO_2_ at 37 °C for 30.0 min. In the final step, 50.0 µL of Assay Buffer B was added, and the plates were centrifuged (800 rpm/RT/2 min). The plates were incubated for 20.0 min, and the fluorescence intensities (ex/em = 485/525 nm and ex/em = 540/590 nm) were measured with use of Tecan Infinite 200 PRO plate reader.

### 4.7. Western Blot Analysis

Equal amounts of proteins after isolation were separated by 7% sodium dodecyl sulfate-polyacrylamide gel electrophoresis (SDS-PAGE), after which the resolved proteins were transferred to a nitrocellulose membrane (Whatman GmbH, Dassel, Germany). The membrane was blocked with 5% milk, followed by exposure to the appropriate antibodies (caspase-3 Antibody Novus 31A11067). β-actin antibody was purchased from Sigma-Aldrich. All bands were visualized using horseradish peroxidase-conjugated secondary antibodies. Secondary antibodies were supplied by Santa Cruz Biotechnology.

### 4.8. Data Analysis

The magnitude of the channel inhibition was calculated as the relative current recorded upon application of the studied compound. The current was defined as I/Icontr; where: I—the peak Kv1.3 current upon an application of an examined compound, Icontr—the peak current recorded on the same cell under control conditions. Inactivation kinetics were fitted by the single exponential function and described by the value of inactivation time constant. In all experiments, statistical analysis was performed by applying the Student’s unpaired *t*-test or one-way ANOVA test. The corrected Bonferroni *p* values were taken into account while performing ANOVA tests. The results were considered statistically significant when *p* < 0.05.

## 5. Conclusions

Simvastatin, mevastatin, and some flavonoids inhibit the activity of Kv1.3 channels in Jurkat T cells. This effect may correlate with the ability of these compounds to induce apoptosis through a reduction in MMP in studied cancer cells. Moreover, the statins may potentiate the biological activity of selected flavonoids. The simultaneous application of these types of chemical compounds could be a promising anticancer strategy. It is suggested that the usage of compounds combination allows for a reduction in the effective dose of therapeutics and thus for a reduction in detrimental side effects in patients.

## Figures and Tables

**Figure 1 molecules-27-03227-f001:**
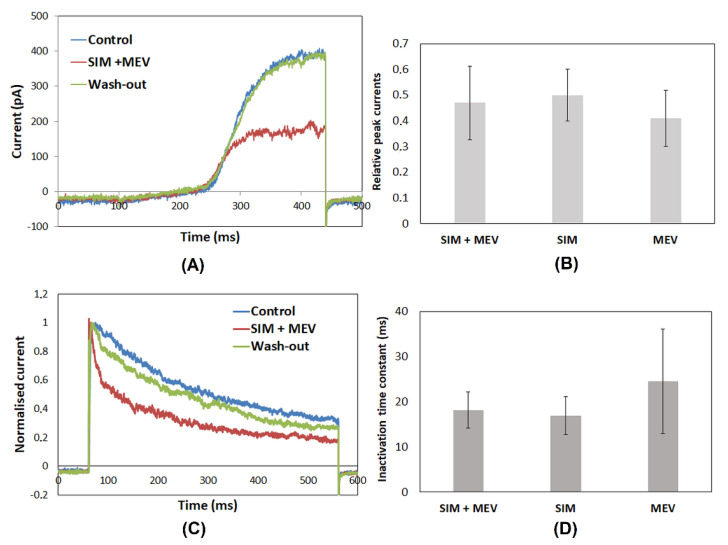
Examples of whole-cell Kv1.3 currents recorded applying the voltage ramp (see Materials and Methods) under control conditions, upon co-application of simvastatin (SIM) at the concentration of 6.0 μM with simvastatin (SIM) at the concentration of 6.0 μM, and after wash-out of the drugs (**A**); relative peak currents (defined in Materials and Methods) upon co-application of the statins and application of each statin alone (**B**); normalized whole-cell Kv1.3 currents recorded applying the voltage step (see Materials and Methods) under control conditions, upon co-application of the statins, and after wash-out of the drugs (**C**); inactivation time constants upon co-application of the statins and application of each statin alone (**D**).

**Figure 2 molecules-27-03227-f002:**
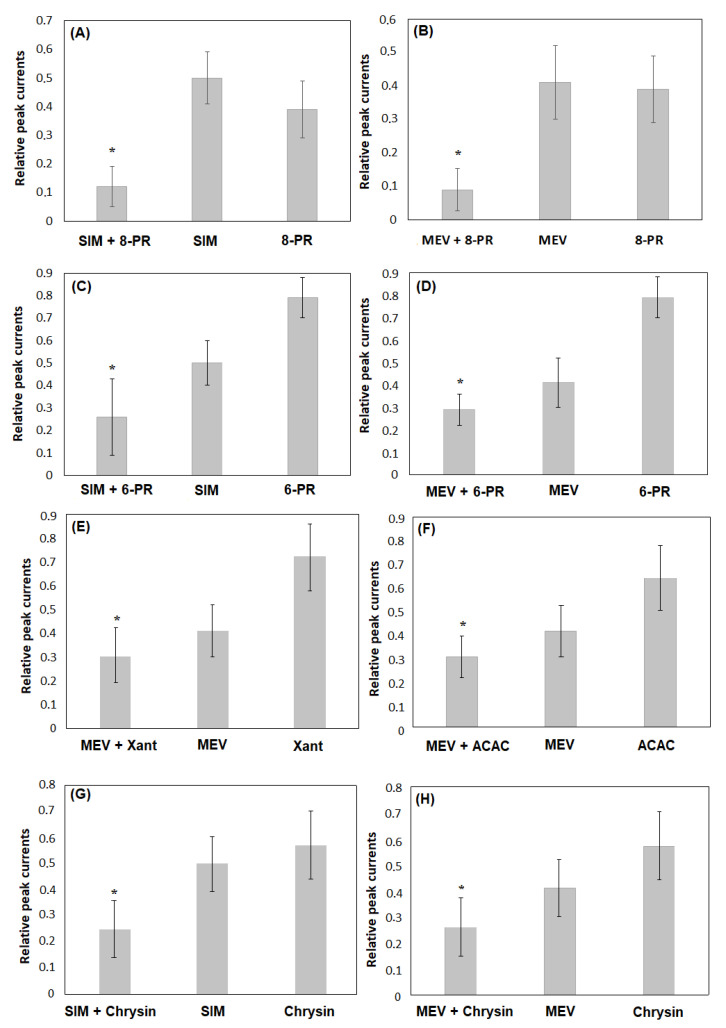
Relative peak currents upon co-application of (**A**) simvastatin (SIM) with 8-prenylnaringenin (8-PR), (**B**) mevastatin (MEV) with 8-PR, (**C**) SIM with xanthohumol (Xant), (**D**) MEV and acacetin (ACAC), (**E**) MEV with Xant, (**F**) MEV with ACAC, (**G**) SIM and chrysin, (**H**) MEV with chrysin, and application of each compound alone; * *p* < 0.05.

**Figure 3 molecules-27-03227-f003:**
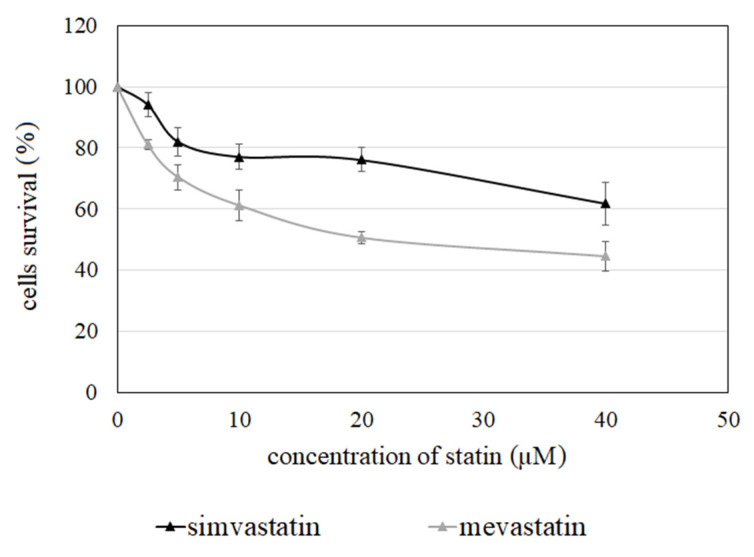
Jurkat T cells survival in the presence of simvastatin and mevastatin.

**Figure 4 molecules-27-03227-f004:**
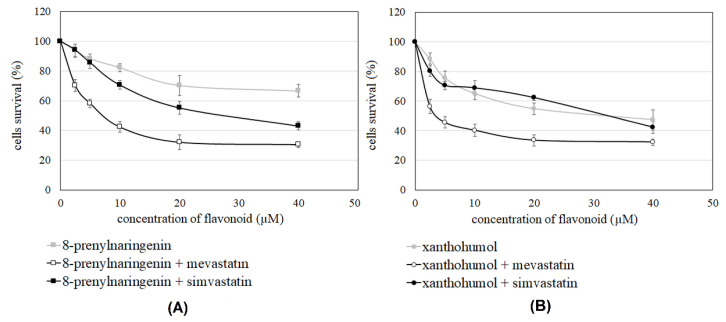
Representative profiles of growth inhibition for pure flavonoids 8-prenylnaringenin (**A**) or xanthohumol (**B**) and their combinations with simvastatin or mevastatin.

**Figure 5 molecules-27-03227-f005:**
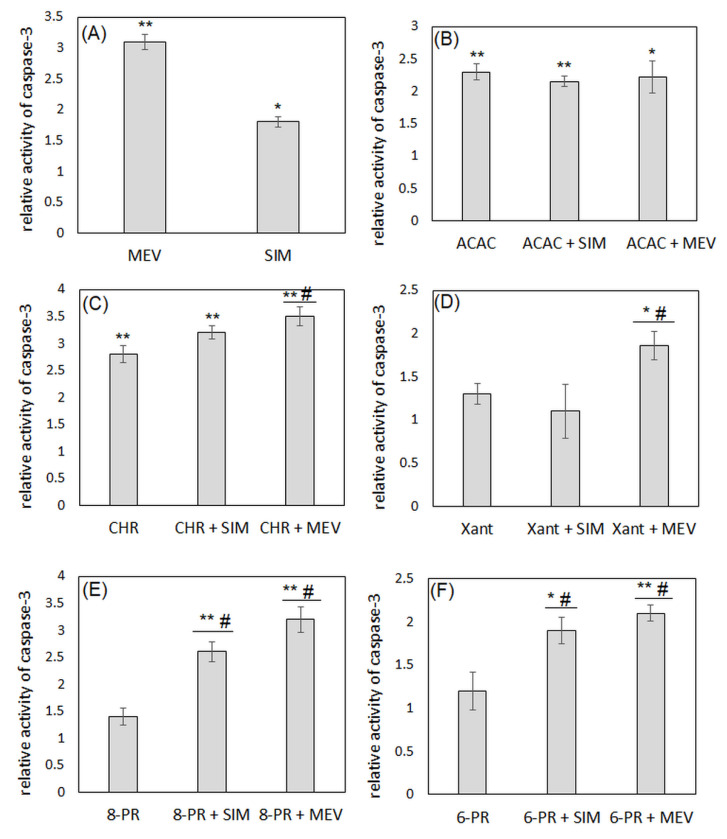
The activity of caspase-3 in Jurkat T cells incubated for 48 h with simvastatin (SIM), mevastatin (MEV) (**A**), flavonoid (acacetin, ACAC; chrysin, CHR; xanthohumol, Xant; 8-prenylnaringenin, 8-PR; 6-prenylnaringenin, 6-PR), and flavonoid in combination with statin (SIM or MEV) (**B**–**F**). Each column represents the mean relative activity of the enzyme normalized to the control derived from non-treated cells. The results were the means ± SDs from three independent experiments. The statistical significance was determined using Student’s *t*-test. We compared the results obtained for each compound/combination with the results obtained from the samples containing no compounds (* *p* < 0.05; ** *p* < 0.001). We also determined the differences between results obtained for single flavonoid and its combination with statins (# *p* < 0.05; # *p* < 0.001).

**Figure 6 molecules-27-03227-f006:**
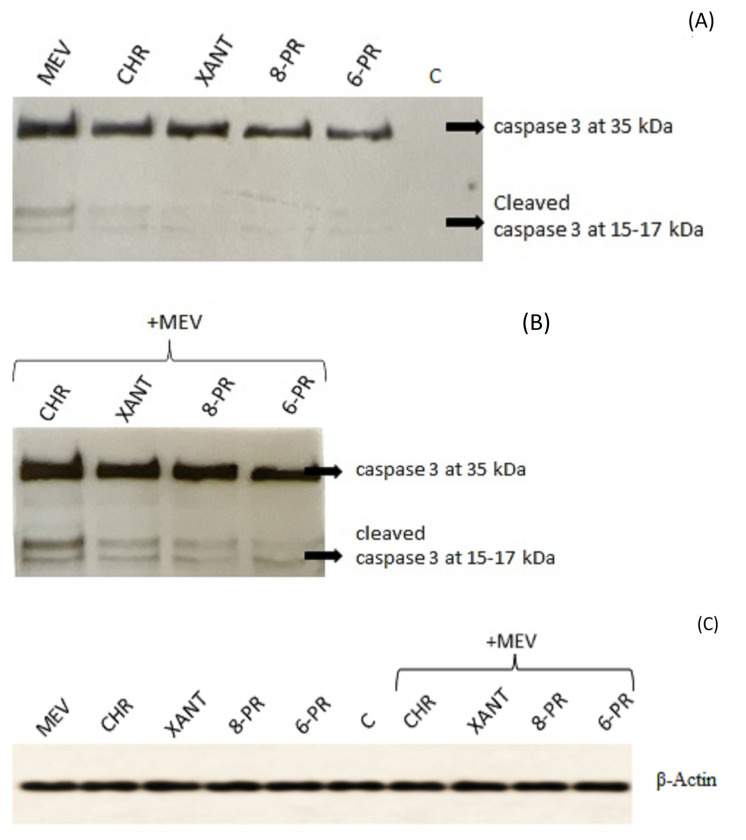
Analysis of caspase-3 protein expression in Jurkat T cells incubated for 48 h with mevastatin (MEV), CHR, XANT, 8 PR, and 6-PR as single agents (**A**), and with their combinations (**B**) for 48 h. Western Blot analysis of Jurkat cell extracts showing full length of cleaved caspase-3 (35 kDa) and cleaved caspase 3 (15–17 kDa); C-untreated cells. The molecular masses of the proteins are indicated on the right side of the gel. β-Actin was used as a reference protein (**C**).

**Figure 7 molecules-27-03227-f007:**
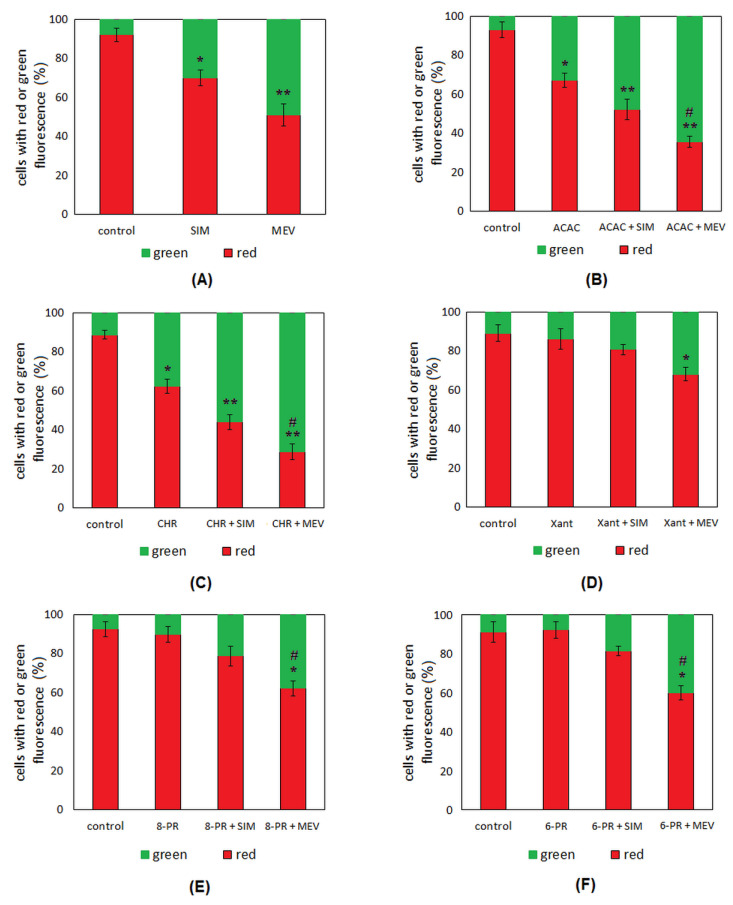
The mitochondrial membrane potential (MMP) of Jurkat T cells incubated for 48 h with simvastatin (SIM) or mevastatin (MEV) (**A**); flavonoid (acacetin, ACAC; chrysin, CHR; xanthohumol, Xant; 8-prenylnaringenin, 8-PR; 6-prenylnaringenin, 6-PR), and flavonoid in combination with statin (SIM or MEV) (**B**–**F**). MMP was expressed as a ratio of 540 nm/590 nm to 490 nm/525 nm fluorescence, as quantified with a fluorescent plate reader after JC-10 staining. Each column represents the percentage of Jurkat T cells characterized by red or green fluorescence. The results were the means ± SDs from three independent experiments. Statistical significance was determined using Student’s *t*-test. We compared the results obtained for each compound/combination with the results obtained from the samples containing no compounds (* *p* < 0.05; ** *p* < 0.001). We also determined the differences between results obtained for single flavonoid and its combination with statins (# *p* < 0.05).

**Figure 8 molecules-27-03227-f008:**
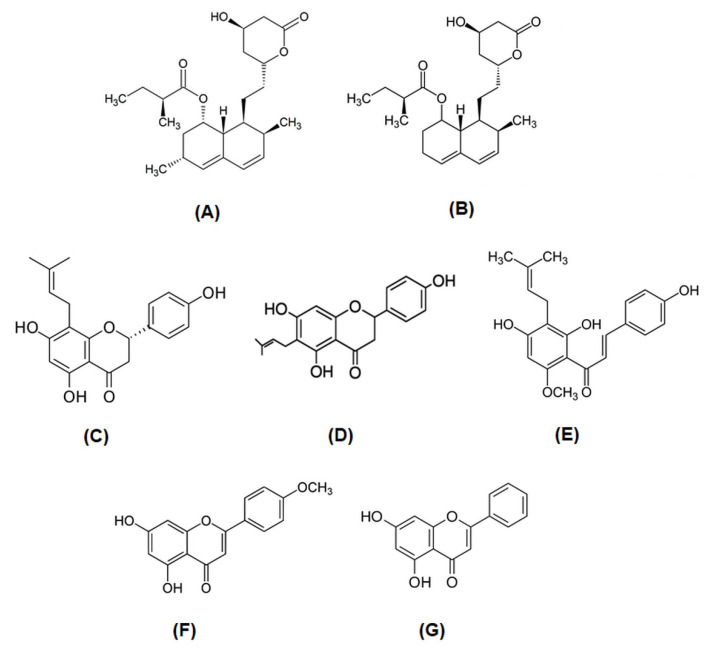
Chemical structures of: simvastatin (**A**), mevastatin (**B**), 8-prenylnaringenin (**C**), 6-prenylnaringenin (**D**), xanthohumol (**E**), acacetin (**F**), and chrysin (**G**).

**Table 1 molecules-27-03227-t001:** Theoretical and experimental values of the relative peak currents upon co-application of simvastatin (SIM) and mevastatin (MEV) with 8-prenylnaringenin (8-PR), 6-prenylnaringenin(6-PR), xanthohumol (Xant), acacetin (ACAC), and chrysin.

SIM Co-Applied with:	8-PR	6-PR	Xant	ACAC	Chrysin
Theoretical values	0.195	0.395	0.36	0.32	0.285
Experimental values	0.12	0.26	0.50	0.53	0.25
Difference	Significant (+)	Significant (+)	Significant (−)	Significant (−)	Not significant
MEV co-applied with:	8-PR	6-PR	Xant	ACAC	Chrysin
Theoretical values	0.16	0.32	0.31	0.26	0.23
Experimental values	0.09	0.29	0.30	0.30	0.26
Difference	Significant (+)	Not significant	Not significant	Not significant	Not significant

**Table 2 molecules-27-03227-t002:** IC_50_ of flavonoids and flavonoids applied with statins (simvastatin or mevastatin) determined in Jurkat T cells.

	No Statin	Simvastatin	Mevastatin
8-prenylnaringenin	n.a.	26.9 µM	7.1 µM
6-prenylnaringenin	n.a. [19]	38.9 µM	34.8 µM
chrysin	26.2 µM [19]	10.8 µM	8.3 µM
xanthohumol	32.5 µM	30.8 µM	3.8 µM
acacetin	n.a. [19]	n.a.	n.a.

n.a.—not available.

## Data Availability

Data are available from the authors A.T., K.Ś.-P. and A.P.-Ł.

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
