# Peer review of "Co-Application of Statin and Flavonoids as an Effective Strategy to Reduce the Activity of Voltage-Gated Potassium Channels Kv1.3 and Induce Apoptosis in Human Leukemic T Cell Line Jurkat"

_molecules, 2022, doi:10.3390/molecules27103227_

Round 1

Reviewer 1 Report

Teisseyre and colleagues in their manuscript submitted to molecules, show that co-application of statins with some flavonoids confer an additive effect in the inhibition of the Kv1.3 channels and induce apoptosis in Jurkat cells.

They measure the effect of co-application of drugs by measuring the membrane potential, cell viability via MTT and apoptosis via caspase-3 activity and loss of MMP, the latter two using commercial kits.

Regarding the graphs showing the membrane potential and inactivation time constant, for sake of simplicity and easiness of reading, I would suggest the authors merge figures 3-7 into a one-page figure and avoid repeating the text exactly 4 times for each combination of drugs. Maybe only highlight the cases in which the effects were statistically significant.

I agree that the use of quantifiable commercial kits for measuring apoptosis might be suitable for this study, but I think the article would benefit from showing cleaved caspase 3 by Western Blot at least in selected, significant combination treatments, as well as Annexin-V staining as an apoptosis measurement. For sake of time, I suggest doing this in combination applications with mevastatin. (Time frame 1-2 months).

Minor comments:

  • recommend to use expression "cancer types" instead of "cancer diseases" or "cancer disorders
  • minor spell check to get rid of a few typos such as "we did not observed" (last d unnecessary) or "ca-application", as well as several double spaces throughout the text
  • Bold table 1 row 5

Author Response

Rewiever #1

  1. It is a lengthy introduction, please reduce the text to half of it. There are too many paragraphs in the introduction section, please combine some of them properly. The reason for the selection of the target compounds is not very clear.

ANSWER: Paper length - the revised version is shortened, as much as possible, in the sections of introduction, results of electrophysiological studies and discussion. The figures 3-7 were reduced to one figure with 8 panels. Only results showing statistically significant differences are depicted in the revised version.

  1. The novelty of present study needs to be clearly introduced. Those tested compounds in this study have already been investigated alone in previous reports, and the only difference is the co-application was employed in this study.

ANSWER: Novelty of the study - is a result of two facts:

  1. a) for the first time, the tested compounds were not applied alone, but in combination
  2. b) for the first time, electrophysiological studies on the selected combinations of compounds were accompanied by a complex non-electrophysiological approach,including cell viability measurements, measurements of apoptosis induction and changes in mitochondrial membrane potential.

Such a complex study had never been performed before in our laboratory.

  1. Choice of the compounds - putative ability to produce additive inhibitory effects when co-applied with each other .
  2. The error bars in the figures (Figure 2-7) are very large, the results are suspicious.

ANSWER: Large error bars - we suspected that the inhibitory effects on the channels were not additive, due to the error bars. Therefore, we performed statistical analysis applying the one-way ANOVA test. The analysis showed that no significant differences occurred in case of Figure 2 (statins co-applied with each other). On the other hand, significant differences (despite large error bars) were obtained for majority of combinations of the statins and the flavonoids (Figures 3-7), one extended Figure 3 in the revised version

  1. It is suggested testing the effects of different combination ratios of compounds.

ANSWER: Experiments with different ratios of the compounds – thank you for this suggestion, it is good idea, however, because of limited time available to prepare revised version, there is no possibility to perform experiments but we think that article would benefit from showing cleaved caspase 3 by Western Blot and these result were added. That was within our capabilities.

Reviewer 2 Report

This study focuses on the influence of co-application of statins (simvastatin or mevastatin) with flavonoids (8-prenylnaringenin, 6-prenylnarigenin, xanthohumol, acacetin or chrysin) on the activity of Kv1.3 channels, viability and apoptosis of cancer cells -human T cell line Jurkat. The manuscript needs to be extensively revised, especially the reduction of the text.

1, It is a lengthy introduction, please reduce the text to half of it. There are too many paragraphs in the introduction section, please combine some of them properly. The reason for the selection of the target compounds is not very clear.

2, The novelty of present study needs to be clearly introduced. Those tested compounds in this study have already been investigated alone in previous reports, and the only difference is the co-application was employed in this study.

3, The error bars in the figures (Figure 2-7) are very large, the results are suspicious.

4, It is suggested testing the effects of different combination ratios of compounds.

5, The “2.1. Electrophysiological studies” can be rewritten, it is a lengthy description.

6, It is a lengthy Discussion section, please make a proper reduction of the text.

Author Response

Rewiever #2

  1. Regarding the graphs showing the membrane potential and inactivation time constant, for sake of simplicity and easiness of reading, I would suggest the authors merge figures 3-7 into a one-page figure and avoid repeating the text exactly 4 times for each combination of drugs. Maybe only highlight the cases in which the effects were statistically significant.

ANSWER: Thank you for this suggestion; Paper length - the revised version is shortened, as much as possible, in the sections of introduction, results of electrophysiological studies and discussion. The figures 3-7 were reduced to one figure with 8 panels. Only results showing statistically significant differences are depicted in the revised version.

  1. I agree that the use of quantifiable commercial kits for measuring apoptosis might be suitable for this study, but I think the article would benefit from showing cleaved caspase 3 by Western Blot at least in selected, significant combination treatments, as well as Annexin-V staining as an apoptosis measurement. For sake of time, I suggest doing this in combination applications with mevastatin. (Time frame 1-2 months).

ANSWER: This is a very valuable comment– thank you for this suggestion. Because of limited time available to prepare revised version, there is no possibility to perform both experiments but we think that article would benefit from showing cleaved caspase 3 by Western Blot and these results were added. That was within our capabilities.

  1. Minor changes suggested by the Reviewer 2 were done.

Round 2

Reviewer 2 Report

This manuscript has been carefully revised, it may be accepted for publication after minor revisions.

1, In the abstract or conclusion, please suggest which co-application is the best for the cancer therapy.

2, In the introduction section, please combine the first three paragraphs into one paragraph.

3, The quality (resolution) of figures can be improved, such as Figure 3.

4, The numerals of the concentrations can be with the same significant digits. For example, “6 μM”, “3 μM”, “30 μM”, and “40 μM” can be changed to “6.0 μM”, “3.0 μM”, “30.0 μM”, and “40.0 μM”, respectively.

5, Figure 4 can be combined into Figure 5. Please modify Figure 6a to the same style as the other graphics.

6, It is suggested to modify Figure 8, the legends indicate “green” and “red”, but they are grey and white in the graphics.

7, Please carefully check the format problems. For example, the “2” in the “CaCl2” and “MgCl2” should be subscript.

Author Response

Rewiever#2

Thank you for your time and valuable comments.

  1. In the abstract or conclusion, please suggest which co-application is the best for the cancer therapy.

ANSWER: Thank you for this suggestion. The sentence “Combinations of simvastatin with chrysin as well as mevastatin with 8-prenylnaringenin seem to be the most promising” was added in the abstract.

  1. In the introduction section, please combine the first three paragraphs into one paragraph.

ANSWER: According to the Rewiever suggestion we combined  the pharagraphs.

  1. The quality (resolution) of figures can be improved, such as Figure 3.

The quality of figures was changed.

  1. The numerals of the concentrations can be with the same significant digits. For example, “6 μM”, “3 μM”, “30 μM”, and “40 μM” can be changed to “6.0 μM”, “3.0 μM”, “30.0 μM”, and “40.0 μM”, respectively.

ANSWER: The numerals of concentrations were changed.

  1. Figure 4 can be combined into Figure 5. Please modify Figure 6a to the same style as the other graphics.

ANSWER: Figure 6 was changed according to the Rewiever suggestion. We decided to not combine Fig 4 and 5.

  1. It is suggested to modify Figure 8, the legends indicate “green” and “red”, but they are grey and white in the graphics.

ANSWER: Figure 8 was modified.

  1. Please carefully check the format problems. For example, the “2” in the “CaCl2” and “MgCl2” should be subscript.

ANSWER: The formatting has been done